# Sustainable Polycaprolactone Polyol-Based Thermoplastic Poly(ester ester) Elastomers Showing Superior Mechanical Properties and Biodegradability

**DOI:** 10.3390/polym15153209

**Published:** 2023-07-28

**Authors:** Jin-Hyeok Choi, Jeong-Jae Woo, Il Kim

**Affiliations:** School of Chemical Engineering, Pusan National University, Busandaehag-ro 63-2, Busan 46241, Republic of Korea; jhchoi95@pusan.ac.kr (J.-H.C.); woojj@pusan.ac.kr (J.-J.W.)

**Keywords:** biodegradation, double metal cyanide catalyst, polycaprolactone polyol, poly(ester ester), thermoplastic elastomer

## Abstract

Thermoplastic elastomers (TPEs) have attracted increasing attention for a wide variety of industrial and biomedical applications owing to their unique properties compared to those of traditional rubbers. To develop high-performance engineering TPEs and reduce the environmental pollution caused by plastic waste, α,ω-hydroxyl-terminated polycaprolactone (PCL) polyols with molecular weights of 1000–4200 g mol^−1^ and polydispersity index (Ð) of 1.30–1.88 are synthesized via the ring-opening polymerization of sustainable ε-caprolactone using a heterogeneous double metal cyanide catalyst. The resulting PCL polyols are employed as soft segments to produce thermoplastic poly(ester ester) elastomers and are compared to conventional thermoplastic poly(ether ester) elastomers prepared from polytetramethylene ether glycol (PTMEG). Notably, the PCL-based TPEs exhibit superior mechanical properties and biodegradability compared to PTMEG-based TPEs owing to their crystallinity and microphase separation behaviors. Accordingly, they have 39.7 MPa ultimate strength and 47.6% biodegradability, which are much higher than those of PTMEG-based TPEs (23.4 MPa ultimate strength and 24.3% biodegradability). The introduction of biodegradable PCLs demonstrates significant potential for producing biodegradable TPEs with better properties than polyether-derived elastomers.

## 1. Introduction

Thermoplastic elastomers (TPEs) possess unique properties, such as high elasticity, high strength, high elastic resistance, low toxicity, wide range of hardness, durable weather and fatigue resistances, and high heat resistance. They are also considered a promising alternative to industrial rubber because of their recyclability [1,2,3,4,5].

TPE is generally composed of thermodynamically incompatible hard segment (HS), soft segment (SS), and low molecular weight (MW) chain extender (CE). In general, HS, which consists of polyester, provides rigidity and strength, whereas SS, which consists of polyether, provides chain flexibility. The thermodynamic incompatibility between HS and SS blocks results in phase separation and formation of microdomains or matrixes. By controlling the morphological structure of these domains, TPEs with desirable properties can be employed for a wide variety of industrial applications, such as elastomers, packaging, smart materials, transportation, and biomedicals [6,7,8,9,10].

A large number of studies have focused on changing the ratio of SS to HS as a means of improving the mechanical and thermal behavior of these materials [11]. In addition, a series of polyether SS and segments of completely different chemical structures have also been investigated [12,13,14,15]. However, studies on polyester-based SSs and their impact on the properties of the corresponding TPEs are surprisingly scarce. Recently, polyester-based TPEs have gained increasing attention because of their biocompatibility and biodegradability. Polyester polyols are commercially produced by the polycondensation reaction of multifunctional aliphatic and aromatic carboxylic acids and polyhydroxyl compounds. They can be further categorized according to their end use. High MW polyols with MWs from 2000 to 10,000 are used to make flexible PUs, while low MW polyols are used for making more rigid products. Conceptually, polyester polyols are readily prepared by condensation polymerization; however, the equilibrium process involves esterification reactions with evolving water, hydrolysis of ester links, and transesterification reactions, and results in a complex mixture of oligomers with a wide range of MWs [16]. Thus, it is an expansive process to produce polyester polyols of a defined structure with a defined MW [16].

Biobased polymers, including biopolymers directly extracted from biomass, such as cellulose, starch, lignin, or bacterial polymers, became a plausible candidate as a sustainable answer to replace fossil-based materials [17,18,19,20]. In addition to reducing the consumption of fossil resources, they reduce the carbon footprint with the decrease of atmospheric CO_2_ emissions. Biobased polymers can be produced by conventional polymerization of biobased building blocks originating from diverse biomass, including agricultural crops, wood and forestry residues, or (micro)algae, and by modification of existing biopolymers.

Polylactones are widely used in specific applications in the human body, such as biomaterials or drug delivery devices [21]. In particular, polycaprolactones (PCLs) are used as internal structure materials, such as artificial bones, because of their excellent mechanical properties and biocompatibility [22]. PCL is a semi-crystalline polymer that can be completely biodegraded within months to years, depending on its MW, crystallinity, intermolecular entanglement, and the decomposition conditions. In general, the amorphous phase is decomposed first, while the MW remains constant and the crystallinity increases. Then, the ester bond is degraded by microorganisms or hydrolysis, resulting in mass loss [23,24,25]. Therefore, PCL polyols with excellent biocompatibility and biodegradability are promising alternatives to conventional petrochemical-based polymers [26,27].

In this study, we developed a protocol for producing biodegradable poly(ester ester) TPEs from PCL, dimethyl terephthalate (DMT), and 1,4-butanediol (BDO). PCL polyols were first obtained via the ring-opening polymerization (ROP) of ε-caprolactone (CL) using heterogeneous double metal cyanide (DMC) catalyst and a bifunctional ethylene glycol (EG) initiator (Figure 1). The resultant PCL polyols and TPEs were characterized by spectroscopic, structural, thermal, and mechanical analyses. The impact of the MW of PCLs and HS/SS ratio on the thermal and mechanical properties and biodegradability of the corresponding TPEs was intensively investigated. A thermoplastic poly(ether ester) elastomer, using polytetramethylene ether glycol (PTMEG) as the SS, was also prepared for comparison.

## 2. Materials and Methods

### 2.1. Materials

CL (99%) and EG (99%) were purchased from Alfa Aesar (Seoul, Republic of Korea). Polymerization grades of DMT and PTMEG were donated by Kolon Industries Co. (Gumi, Republic of Korea) and used after drying. Irganox 1010 and titanium tetrabutoxide (TBT; 97%) were purchased from Sigma-Aldrich (Seoul, Republic of Korea). Reagent grade chloroform, dichloromethane, diethyl ether, ethanol, hexane, and phenol were purchased from Dae Jung Chemical Co. (Daejeon, Republic of Korea) and distilled prior to use. Trifluoracetic acid (TFA; 99%) and 1,1,2,2-tetrachloroethane (97%) were purchased from Tokyo Chemical Industries (Tokyo, Japan) and used without further purification. The Zn/Co DMC catalyst was prepared using diethyl malonate (DEM) as a complexing agent and Pluronic P123 ((EO)_20_(PO)_70_(EO)_20_) [PEO-*b*-PPO-*b*-PEO copolymer, PEO = poly(ethylene oxide), PPO = poly(propylene oxide)] as a co-complexing agent following a previously reported procedure [28]. The experimental formula of the catalyst is Zn_1_._99_Co(CN)_6_._25_ ·0.51DEM·0.03P123·1.63H_2_O·1.47Cl^−^.

### 2.2. Synthetic Procedures

#### 2.2.1. Preparation of Polycaprolactone Diols

PCL diols were synthesized via the ROP of CL using a DMC catalyst [29,30]. Typically, CL polymerization is conducted in a 250 mL two-neck round-bottom flask with a magnetic stir bar under a nitrogen atmosphere. Initially, the flask was charged with an EG initiator and DMC catalyst. After purging with nitrogen for 30 min, CL was added and allowed to react at 160 °C for 8–20 h depending on the targeted MW. The crude reaction mixture was dissolved in chloroform and filtered to remove the DMC catalyst. The filtrate was precipitated in diethyl ether and vacuum-dried to a constant weight to obtain a PCL polyol.

#### 2.2.2. Synthesis of Thermoplastic Poly(ester ester) Elastomers Using PCL Diols

TPEs were synthesized in a 600 mL high-pressure stainless-steel reactor (Parr Instrument Co., Moline, IL, USA) equipped with a vacuum pump and Dean–Stark trap to collect by-products, such as methanol and 1,4-BDO [31,32]. TBT (0.3 wt% with respect to DMT) and Irganox^®^ 1010 (0.5 wt% with respect to the total mass of monomers) were used as a catalyst and thermal stabilizer, respectively. First, 10 g of DMT and 0.1 g of Irganox^®^ 1010 were added to the reactor under nitrogen flow. Then, 6.96 g of BDO (molar ratio of DMT to BDO was maintained at 1:1.5) was added at 70 °C and stirred for 20 min. After that, 3 mg of TBT in 1 mL of hexane was added at 90 °C and the transesterification reaction was carried out under nitrogen atmosphere at 160–210 °C for 90 min. The conversion of the transesterification reaction can be measured using the amount of methanol by-product. After the cessation of methanol distillation, 10 g of PCL polyol was added to the reactor, and the polycondensation reaction was conducted at 250–255 °C under vacuum. The BDO by-product was also distilled off at this stage (see Figure 1). The change in torque (during stirring) was monitored to estimate the melt viscosity of the product. The polycondensation reaction was allowed to react for 2–4 h. The resulting TPEs were dissolved in chloroform, purified by precipitation from excess diethyl ether, washed with ethanol, and vacuum dried at 80 °C for 24 h.

### 2.3. Biodegradable Test of TPEs

Biodegradation tests under aerobic conditions were conducted to measure the degree of aerobic biodegradability of the plastic materials in accordance with ISO-14852 [33]. In summary, the aerobic state was maintained at a rate of 100 mL min^−1^ through flow controllers. Air passed through 500 mL of 10 M KOH to remove the remaining CO_2_ from the air. The air then passed through 0.0125 M 100 mL Ba(OH)_2_, and the remaining CO_2_ was determined through the turbidity of Ba(OH)_2_. In the flask bearing the TPE sample, there were various nutritional media, compost, and 1 L of distilled water where CO_2_ was removed from the atmosphere. The TPE sample was placed in a degradation test flask, and the biodegradability experiment was conducted for 3 months. As a result, in a flask containing 200 mL of 0.0125 M Ba(OH)_2_, CO_2_ was generated by the respiration of microorganisms and biodegradation of the sample by microorganisms. CO_2_ reacted with Ba(OH)_2_ to form BaCO_3_, and the remaining Ba(OH)_2_ was titrated using 0.05 M HCl to obtain the generated CO_2_. The amount of CO_2_ generated through biodegradation was determined by subtracting the CO_2_ generated in the blank test from the titration result value of CO_2_.

### 2.4. Characterization

The intrinsic viscosity (*η*) of the TPEs was analyzed at 30 °C using a capillary Ubbelohde viscometer (Dong-Jin Intrument Co., Seoul, Korea). The concentration of TPEs was maintained at 0.5 g dL^−1^ in a mixture of 1,1,2,2-tetrachloroethane and phenol (40:60 *v/v*). The viscosity average MW (*M_v_*) of the TPEs was analyzed using the Mark−Houwink equation (*η* = K*M_v_* ^a^, where K = 5.36 × 10^−4^ and a = 0.697) [31,34]. ^1^H (400 MHz) NMR spectra of PCL diols and TPEs were measured using a Varian INOVA 400 NMR spectrometer (Palo Alto, CA, USA). The MW and polydispersity index (*Ð*) were measured by gel permeation chromatography (GPC) using a Waters 150 instrument (Waters Korea, Seoul, Korea) operating at 40 °C with 104, 103, and 500 Å columns in tetrahydrofuran (THF). Low *Ð* poly(ethylene glycol) standards were used for calibration. The glass transition (*T_g_*), crystallization (*T_c_*), and melting (*T_m_*) temperatures were determined using differential scanning calorimetry (DSC; Q100 instrument, TA Instruments, New Castle, DE, USA) in the range of −80 °C to 240 °C and at a heating and cooling rate of 10 °C min^−1^. The *T_g_* of the TPEs was determined as the temperature at the maximum relaxation peak of the loss modulus (*E*″) and tangent delta (tan *δ*). Thermogravimetric analysis (TGA) was performed using a TGA Q50 analyzer (TA Instruments) from 30–800 °C at a heating rate of 10 °C min^−1^. Dynamic mechanical analysis (DMA) (Q800 V21.2 Build 88) measurements were performed using a dynamic mechanical thermal analyzer (TA Instruments) operating in tension mode at a frequency of 1 Hz. The samples were heated at a rate of 5 °C min^−1^. The rubbery plateau was determined using the storage modulus (*E*′). Wide-angle X-ray scattering (WAXS) measurements of the TPEs were performed using an X’Pert3 MRD with computerized data collection and analytical tools. The WAXS curves of the PCL-based TPE samples were measured in the 2θ range of 5–80° with a step size of 0.05°. The X-ray source, which was a Cu Kα radiation of wavelength *λ* = 1.54 Å, was formulated using an applied voltage of 40 kV and a filament current of 30 mA. The stress–strain curves of the PCL-based TPE samples were measured using a universal testing machine (KSU05, Kyungsung Testing Machine Co., Ansan, Republic of Korea) and a constant crosshead speed was maintained at 100 mm min^−1^. The uniformly shaped specimens were analyzed at 25 °C. A minimum of three specimens were tested for each value. The elongation percentage and ultimate strength of the TPE samples were analyzed with the stress–strain curves. The hardness of the PCL-based TPEs was determined using a Durometer Shore A type and Shore D type (GS-706N, GS-702N, Teclock, Nagano, Japan). The TPE film and electrospun fiber morphologies were investigated in order to examine the effectiveness of biodegradability using scanning electron microscopy (SEM; Hitachi S-3000H, Tokyo, Japan). Before the analysis, the samples were fixed to copper stubs with carbon adhesive tape and sputter coated with 10 nm gold particles. The functionality of polyols was measured using a potentiometric titrator (888 Titrando, Methrohm, Singapore). The hydroxyl number was calculated using the first and second potentiometric titration value measurement endpoints of 0.5–1 and 5–10 mL, respectively. Absolute MWs of PCL diols were also measured by matrix-assisted laser desorption ionization–time of flight mass spectrometry (MALDI-TOF MS) using a Waters Maldi Synapt G2-Si Mass Spectrometry (Waltham, MA, USA). Samples were dissolved in THF, and the absolute MW was measured in the range of 500–10000 g mol^−1^ in linear mode through a matrix of 2,5-dihydroxybenzoic acid.

### 2.5. Mesoscale Simulation Method

Mesoscale simulation of TPEs was performed using the MesoDyn module in a commercial software Materials Studio^®^ 4.2 (Accelrys, Inc., San Diego, CA, USA) [35], successfully employed to investigate several polymer systems [36,37,38] The MesoDyn module is a dynamic variant of mean-field density functional theory, providing a coarse-grained method for the study of complex fluids, their kinetics, and their equilibrium structures at large length and time scales. MesoDyn can analyze these morphologies both quantitatively and qualitatively. Before employing the MesoDyn simulation, the components of TPEs were represented by coarse-grained models. The interaction between HS beads consisting of DMT and BDO segments and PCL SS or PTMEG SS beads were represented by Flory–Huggins interaction parameters (χ). The solubility parameters were obtained by using quantitative structure–property relationship (QSPR) methods that are available in the Synthia module in Materials Studio^®^. The effective χ between various beads used in this work are listed in Table 1. In the MesoDyn simulation, the dimensionless parameters were chosen as follows: the size of cubic grid 64 × 64 × 64 nm, the bond length 1.1543, the bead diffusion coefficient 10^−7^ cm^2^ s^−1^, the noise parameter 100.0, the compressibility parameter 10 kT, the simulated temperature 298 K, and the time step 0.5 ns. For each system, a total number of 20,000 steps was carried out to reach a kinetic equilibrium.

## 3. Results and Discussion

### 3.1. Preparation of Polycaprolactone Diols with Different Molecular Weights

The bulk ROP of CL was performed at 160 °C using various molar ratios of CL to EG ([CL]/[EG] = 9, 17, 22, 26, 30, and 35) to produce PCLs with different MWs. Figure 2 shows ^1^H NMR spectra of the reaction mixtures collected at different times. The results clearly show the intensity of the peak assigned to α-H in CL decreases according to polymerization time, while the peak assigned to α-H in PCL [29,39] increases accordingly. At a high concentration of initiator, say [CL]/[EG] = 9, monomer conversion was higher than 95% after 8 h of polymerization; however, it took longer time to achieve the similar level of conversion at a low concentration of initiator, say [CL]/[EG] = 35. Figure 2G illustrates conversion versus time plots collected at different [CL]/[EG] ratios.

Independent polymerization at 160 °C was performed in the absence of the EG initiator, showing that it took more than 48 h to achieve 50% conversion. In addition, taking a careful look at the progress of polymerization at [CL]/[EG] = 9 (Figure 2A), the proton peaks in EG disappear at the early period of polymerization by transfer reactions. These results clearly show that the EG initiator acts not only as a transfer reagent but also as an activator of ring-opening of CL coordinated to Zn active sites. Figure 2H shows GPC curves of the resultant PCLs obtained at various molar ratios of CL to EG, and the MW and polydispersity index (*Ð*) values are summarized in Table 2. The theoretical MW simply calculated by [CL]/[EG] ratio is well in line with the measured number average molecular weight (*M_n_*) value, again demonstrating the EG initiator works efficiently as a chain transfer agent. As the MW increases, considering bulk polymerization, the free movement of growing chains must be limited, resulting in somewhat large *Ð* values (1.8 at [CL]/[EG] = 35). GPC curves contain tails indicating small MW moieties. The hydroxyl functionalities of the resultant polymers around 2.0 again demonstrate the efficiency of the EG initiator as a transfer agent. The considerable deviation of the functionality from desirable 2.0 of the PCL4000 sample comes from the limitation of movement of the growing polymer chains due to the increase of medium viscosity induced by physical entanglement.

A critical limitation to developments in polymer end-group functionalization lies in the difficulty in confirming end-group purity. Modern characterization techniques, especially MALDI-TOF MS, have proved invaluable in confirming end-group fidelity since MALDI-TOF is a soft ionization technique that enables resolution of individual n-mers of polymers in a mass spectrum distribution. This resolution enables the elucidation of not only mass distribution and repeat unit mass but also the identity and fidelity of end groups. Figure 2I shows that the PCL3000 sample has a narrow, monomodal distribution. The molecular weight distributions for this polymer were calculated to be *M_n_* = 3508.7 and *M_w_* = 3824.5, leading to a narrow distribution with *Đ* = 1.09.

The peaks within the spectrum correspond to a single distribution, with each peak separated by 114.13 mass units, suggesting that only one set of end groups exists within this sample. The end groups can be verified by looking at individual n-mers, such as the 30-mer [inset in Figure 2I]. The theoretical mass value of the 30-mer with two hydroxyl end groups is 3508.9272, which was calculated by multiplying the repeat unit m/z (114.1293) by the number of repeat units (30), adding the m/z of the EG initiator (62.0584), and finally adding the m/z of the sodium cation (22.9898). The observed value for the 30-mer is 3508.6, which is about 0.3 Da different from the theoretical value. The data collected from MALDI-TOF MS again demonstrate that the EG initiator works well to form caprolactone ended with two hydroxyl groups. Note that there are small peaks beside main PCL diol peaks. For example, the observed values for the 30-mer are 3508.6, corresponding to the m/z of [EG + 30(CL) + Na+], and 3463.7, corresponding to the m/z of [HO + 30(CL) + Na+; theoretically 3463.8756], indicating that trace amount of moisture existing in the polymerization system works as a transfer agent to form carboxylic acid end group at one end and hydroxyl end group at the other end. The ratio of EG-initiated PCL to H_2_O-initiated PCL is 98.0 to 2.0. The measured Mn value of 3508.6 from MALDI-TOF MS is larger than that of 3000 from GPC, and there is no shoulder peak of small MW moieties in MALDI-TOF MS chromatogram. Considering GPC curves were obtained using polymerization mixtures and MALDI-TOF MS analysis were based on the purified samples, the impurity consisting of small MW moieties was easily removed by simple purification.

Based on the results collected from the kinetics of polymerizations using NMR measurements, GPC, and MALDI-TOF MS analyses, a plausible mechanistic pathway of DMC-catalyzed ROP of CL is initiated by EG (Figure 3). The nucleophilic attack of the Zn-coordinated EG initiator to the CL monomers coordinated to the same and/or neighboring Zn sites helps the fast ring-opening of CL monomers. This transfer reaction results in hydroxyl groups at the PCL chain ends. These hydroxyl groups will play roles as nucleophilic attackers to induce the ring-opening of CL. Repetition of these procedures results in PCL diols with uniform structure. On the other hand, PCL with a hydroxyl group at one end and a carboxylic acid group at the other end is obtained if a small amount of water molecules remains in the polymerization system and works as initiators.

The glass transition temperature (*T_g_*) of the PCL diol samples varied from –60.0 °C to –65.4 °C with no specific trend according to MW, as shown in Table 2. Interestingly the double melting peak was observed on the heat-flow curve for all samples. DSC heat/cool/heat experiments are designed to erase previous thermal history by heating each sample above a melting transition, where relaxation or molecular rearrangement can occur, then cooling at different rates (10.0 °C/min and 100 °C/min) before heating again. Regardless of the cooling rate difference, a double melting peak was observed. The double endothermic behavior is due to the existence of two kinds of crystal lamellae with different thicknesses [40,41]. So to speak, the lower temperature endotherm comes from the melting of the thinner lamellae, and the higher temperature endotherm comes from the thicker lamellae. A different explanation can be given. The lower temperature endotherm is also caused by the melting of the lamellae initially present. The partially melted PCL sample may recrystallize into a more perfect lamellae, resulting in melting at higher temperatures [42,43]. The existence of one exotherm at practically more similar temperatures than the endotherms indicates that as the cold crystallized PCL melts, the molecules arrange in more perfect lamellae.

### 3.2. Synthesis of Thermoplastic Poly(ester ester) Elastomers

The HSs of the TPEs were first prepared via the transesterification of DMT and BDO using a TBT catalyst. Then, they were reacted with the α,ω-hydroxyl-terminated PCL with MWs of 1000–4000 g mol^−1^, PCL1000, PCL2000, PCL2500, PCL3000, PCL3500, and PCL4000, to produce desired TPEs (see Figure 1). The representative poly(ether ester) elastomer was also prepared using PTMEG as the SS for comparison. The ratio of HS to SS was 50/50 *w/w*, and it was modulated to prepare TPEs with 40–60% of HS in the case of PCL3000. ^1^H NMR spectra of TPEs prepared using PCL3000 in different HS/SS ratios are shown in Figure 4 as representative examples. Signals attributed to the aromatic protons of DMT are located at 8.08 ppm, and signals attributed to the methylene protons of BDO, namely CH_2_−CH_2_−CH_2_−CH_2_ and CH_2_−CH_2_−CH_2_−CH_2_, are located at 1.96 and 4.42 ppm. The signals corresponding to the methylene protons of the PCL segment appeared at 4.33, 4.05, 2.32, 1.62, and 1.47 ppm. Integration of the peaks a, b, and g in Figure 4 made it possible to calculate HS to SS ratios. Table 3 summarizes the results of polycondensation. Feeding HS/SS ratio as 50/50 using PCL1000, PCL2000, PCL2500, PCL3000, PCL3500, and PCL4000 yielded poly(ester ester) elastomers of HS/SS ratios of 40/60, 44/56, 39/61, 41/59, 40/60, and 47/53. Note that the HS/SS ratio of PTMEG-based poly(ether ester) elastomer was 45/55. Modulating HS/SS ratios as 40/60 and 60/40 using PCL3000 yielded poly(ester ester) elastomers of HS/SS ratios of 27/73 and 57/43.

The intrinsic viscosity (*η*) values of the poly(ester ester) elastomers are in the range of 1.43–2.35 dL g^−1^, which corresponds to the *M_v_* values in the range of 92,000–189,000 g mol^−1^ [31,34]. These results demonstrate that high-MW poly(ester ester) elastomers were successfully generated and that the changes and conspicuous trends corresponding to compositional changes induced by the modification of the reaction parameters are not significant. Considering the measured HS/SS ratios and the *M_n_* values of poly(ester ester) elastomers prepared using PCL3000, the calculated compositions of poly(ester ester) elastomers of HS/SS ratios of 27/73, 41/59, and 57/43 are poly[(HS)_5_-(SS)_42_]_11_, poly[(HS)_9_-(SS)_42_]_9_, and poly[(HS)_18_-(SS)_42_]_7_, respectively.

### 3.3. Thermal Properties of TPEs

The properties of TPEs are generally attributed to their microphase-separated morphology, which consist of microdomains rich in HSs and a microphase rich in SSs that arise from the thermodynamic immiscibility of the HSs and SSs. The degree of incompatibility is largely determined to a great extent by the ratio of intra- and intermolecular interactions. TPEs have an SS microphase region (*T_g_*_1_), blended SS and HS microphase (*T_g_*_2_), and HS microphase region (*T_m_*) [12].

The *T*_g_ and *T_m_* values of TPEs were determined by DSC in the range of −45 to 240 °C (Figure 5A,B). The *T_g_* value obtained for PTMEG-based poly(ether ester) elastomer (−22.9 °C) was lower than that obtained for poly(ester ester) elastomers (−11.1 °C to −14.7 °C) synthesized at the same HS/SS ratio. This is likely because of the more flexible structure of PTMEG compared to PCL segments [44,45]. Because the PTMEG has a relatively linear structure with a low free volume, crystal formation is easy and microphase separation clearly occurs. In addition, phase separation occurs more easily because the compatibility between hard phase and PTMEG is low. Accordingly, it is more easily crystallized in the hard phase. On the other hand, PCL has a relatively nonlinear structure and high free volume, making it relatively difficult to form crystals. As a result, the *T_m_* values of the PCL-based poly(ester ester) elastomers are relatively low [45]. The changes in *T_g_* values have no conspicuous trend according to the MW of PCL, while *T_m_* values decrease monotonously according to the MW of PCL as expected. Thus, the *T_m_* value of 5/5-PCL1000 sample, 120.5 °C, decrease to 113.0 °C for 5/5-PCL4000 sample. The effect of soft segment to hard segment ratio on the thermal property of poly(ester ester) elastomers using PCL3000 as a soft segment. The *T_g_* and *T_m_* values increased with increasing hard segment content. Accordingly, the *T_g_* and *T_m_* values decreased in the order of 6/4-PCL3000 (−0.51 °C and 135.48 °C, respectively) > 5/5-PCL3000 (−11.57 °C and 116.58 °C, respectively) > 4/6-PCL3000 (−21.09 °C and 83.25 °C, respectively). In TPEs, *T_g_* of the soft segments generally increases as part of the hard segments are dissolved into the soft domains due to phase mixing. Since the *T_g_* of pure PCL is of about –60 °C [45], the variation of *T_g_* value suggests that the dissolution of hard segments into PCL domains is considerable, especially at higher hard segment composition.

The viscoelastic behavior of the TPE films induced by microphase separation was investigated using DMA analysis. Evolution of the elastic (storage) modulus (*E*′), connected with the rigidity of the sample, the viscous (loss) modulus (*E*″), correlated with the flowing properties, and the loss factor (damping) (tan *δ*), represented by the ratio between the dissipated energy and the elastically stored energy, with temperature for TPEs are illustrated in Figure 6. In the case of the PTMEG-based TPE (5/5-PTMEG), there are four distinct regions according to its molecular mobility: sub-glass transition (glassy) region at approximately −70 °C, glass transition region (from −70 °C to 30 °C), rubbery plateau (from 30 °C to 150 °C) that is influenced by the MW, intermolecular entanglement, crystallinity, crosslinking and secondary bonding, and flowing region (above 170 °C). Thus, the 5/5-PTMEG sample denotes the pattern of a well-behaved elastomer. The PCL-based TPE samples show quite complex viscoelastic behaviors. Not every peak recorded on the evolution of *E*′, *E*″, and tan *δ* with temperature can be correlated with a relaxation. Overlapping may result in complicated patterns of the viscoelastic behaviors versus temperature. Samples based on PCL seem to be one of these cases. On the other hand, the decrease in *E*′ observed for PCL-based TPEs occurs at a higher temperature (−15 to −10 °C). This is likely because of a stronger secondary bond at a lower temperature of PCL SS compared to that of the PTMEG SS [46]. The flat area on the rubber phase is negligible at 0–130 °C, and *E*′ continuously decreases. The lower degree of microphase separation may be due to the better compatibility between PCL SSs and ester groups of the HSs. In addition, a large area is associated with the blended HSs and SSs. Owing to less crystal formation, there is a negligible flat area on the rubber phase [45].

As shown in Figure 6C (tan *δ*), most of the samples represent four specific areas [47]. For 5/5-PTMEG1000, a convolution appears with an increase in temperature because of the amorphous region of PTMEG and amorphous part of the HSs from −100 to −80 °C. In the case of PCL-based TPEs, this section appears over a wider area, namely from −100 to −50 °C. Next, the peak at which tan δ increases is the same as *T*_g_, as shown in the mitigation process of E′ (Figure 6A). These signals indicate that glass transition occurs in the amorphous regions of the SSs and HSs, resulting in an increase in *E*″ and a decrease in tan *δ*. *T_g_*_2_ appears, and tan δ increases in the region where HSs and SSs are blended. In this region, 5/5-PTMEG1000 exhibits a low *T_g_*_2_ value at approximately 50 °C. In the case of PCL-based TPEs, *T_g_*_2_ signals are clearly observed over a wide area owing to the high compatibility between PCL SSs and DMT-based HSs. Thus, HSs and SSs are widely blended. The increase in tan *δ* after the appearance of the *T_g_*_2_ signals indicates the melting of the HSs crystal phase. Owing to the high degree of microphase separation in 5/5-PTMEG1000, the tan *δ* value is maintained behind the glass transition region, whereas in the case of PCL-based TPEs, tan *δ* continuously increases because of the wide HS/SS blended area. The differences in tan *δ* values according to the HS/SS ratios were investigated for 4/6-, 5/5-, and 6/4-PCL3000 (Figure 6F). As the SS content increases, the glass transition region tends to appear at lower temperatures. 6/4-PCL3000, with the lowest SS content, exhibits a behavior similar to that of PTMEG because it has high crystallinity owing to the high HS content [36,45].

### 3.4. Tesile Properties of Thermoplastic Poly(Ester-Ester) Elastomers

The mechanical properties are related to the elongation percentage, ultimate strength, and Young’s modulus. The stress–strain curves of the PTMEG-based and PCL-based TPEs are shown in Figure 7, and the results are summarized in Table 4. The PCL-based TPEs wherein PCL1000 was used as an SS, namely 5/5-PCL1000, exhibited similar elongation (1105%) but significantly enhanced ultimate strength (33.83 MPa) compared to 5/5-PTMEG1000 (1166% and 23.4 MPa, respectively). As mentioned in the thermal analysis in Section 3.3, the sum of the stress exhibited by the low crystallinity of the HS region and the low but wide region of HS/SS of 5/5-PCL1000 was greater than the high crystallinity of the HS region of 5/5-PTMEG1000 at room temperature. Thus, the enhanced mechanical properties of PCL-based TPEs depend on the MW of the PCLs [48]. A slight increase in the ultimate strength was observed when the MW of PCL was increased from 1000 to 3000 g mol^−1^. Among the TPEs, 5/5-PCL3000 exhibited the highest ultimate strength (39.72 MPa), and a significant decrease in the stress and strain was observed for 5/5-PTMEG4000. This is due to the relatively low functionality of PCL4000. Therefore, the MW was insufficient for the synthesis of a TPE. The ultimate strength values also increased with an increase in DMT content, as observed for 4/6-, 5/5-, and 6/4-PCL3000 (33.96, 39.72, and 44.92 MPa, respectively), whereas elongation values decreased from 1638 to 820%.

The Shore hardness of TPEs was measured according to ISO 868. The Shore A and D hardness values of PTMEG-based poly(ether ester) elastomer are 85 and 29, respectively. The Shore A and D hardness values of PCL-based poly(ester ester) elastomers range from 90 to 95 and 35 to 40 for the HS/SS of 50/50 samples, respectively, demonstrating PCL-based TPEs are harder than PTMEG-based TPEs so that they are rigid and cannot be deformed by hand. The hardness tended to increase with increasing HS/SS ratios, as expected.

### 3.5. WAXD Analysis of Thermoplastic Poly(Ester-Ester) Elastomers

The crystalline structure and microphase separation morphology of TPEs were examined using WAXD. Figure 8 shows the WAXD patterns of TPEs after annealing the sample films at 80 °C for 24 h. Both PTMEG- and PCL-based TPEs show reflections at 2θ = 16.0, 17.2, 20.7, 22.3, 23.2, and 25.3° (*d*-spacing = 5.5, 5.2, 4.4, 4.0, 3.8, and 3.5, respectively), which correspond to the characteristic diffractions of α-form holocrystalline [31,49]. As mentioned in the thermal analysis at Section 3.3, 5/5-PTMEG1000 shows high crystallinity and, thus, its WAXD pattern exhibits a higher intensity than that of the PCL-based TPEs. As the HS content increased, the intensity of the scattering peaks also increased owing to the increase in crystallinity (Figure 8B).

Since the large size and slow dynamics of polymer structure makes quantum mechanics or molecular modeling with atomistic detail prohibitively expensive, the MesoDyn module in Materials Studio^®^ was employed to elucidate the mesophase behavior of the TPEs. At such length scales, fast atomic degrees of freedom only contribute to effective potentials. The strength of MesoDyn is that full simulations are performed on systems in which the chemical nature of the species is included. The chemistry of the system is imparted through the molecular architecture and the bead–bead repulsions. In order to determine MesoDyn input parameters for systems of poly(ester ester) TPEs consisting of PCL soft segments and DMT/BDO hard segments and poly(ether ester) TPEs consisting of PTMEG soft segments and the same hard segments, we assumed the TPEs have a molecular weight of 40,000. The Flory–Huggins interaction parameters must be obtained for each pair of these. The solubility parameters were obtained by using quantitative structure property relationship (QSPR) methods that are available in the Synthia module in Materials Studio^®^. The solubility parameters were used to derive Flory–Huggins interaction value (*χ*) using:(1)$$\chi =\frac{V_{ref}{\left(\delta_{a}-\delta_{b}\right)}^{2}}{RT}$$
where *V_ref_* is a reference volume, taken to be the molar volume of one of the monomers, *δ_a_* and *δ_b_* are solubility parameters, taken to be the van Krevelen solubility parameter, and *T* is temperature. Converting these parameters into the MesoDyn input parameter using the characteristic ratios of each segment of TPEs through:(2)$$v^{-1}\varepsilon_{ab}=\chi_{ab}RT$$
where *ν*^−1^*ε_ab_* is the input parameter.

Figure 9 displays the simulation cell and the associated density field distribution. The HS blocks (A) consisting of DMT, BDO, and PTMEG (C) or PCL (B) SS blocks are incompatible, and phase separations clearly occur, which can be clearly observed from both density field distributions and beads isosurface representations. Comparing (A_1_-*b*-B_2_)_40_ poly(ester ester) TPE with (A_1_-*b*-C_2_)_40_ poly(ether ester) TPE, no mentionable differences are found. In the poly(ester ester) TPE with the same SS and HS compositions, beads isosurface representations clearly show the phase diagrams become more complicated, and the domain size tends to become smaller as the MW of PCL SS increases. The morphology of (A_1_-*b*-B_2_)_40_ is quite different from that of (A_2_-*b*-B_4_)_20_; however, no conspicuous differences are observed among (A_2_-*b*-B_4_)_20_, (A_3_-*b*-B_6_)_15_, and (A_4_-*b*-B_8_)_10_.

The modulation of HS to SS ratio by increasing the size of HS block (A) from (A_2_-*b*-B_4_)_20_, (A_3_-*b*-B_4_)_20_, to (A_4_-*b*-B_4_)_20_, a dramatic change of morphology is observed (Figure 9D–F). As the HS segment increases, the density fields of HS and SS become closer and overlap each other, demonstrating the compatibilization of both segments. Considering the lack of direct inclusion of electrostatics in MesoDyn simulation is a critical limitation, including no long-ranged potentials for computational efficiency, it may be impossible to directly match the experimental results with simulation results. Thus, the rapid algorithm used in this study allows only a rough and speedy estimation of a pseudo-dynamic equilibrium morphology. Simulations with the full potential and density field solver space are needed to obtain more accurate results.

### 3.6. Biodegradable Properties of Thermoplastic Poly(ester ester) Elastomers

#### 3.6.1. Hydrolytic Degradation Tests

A hydrolysis experiment was conducted using 2 × 2 cm^2^ TPE films at 45 °C in an aqueous 3% NaOH solution. The degree of hydrolysis was calculated using Equation (3) [50]:(3)$$Weight\;loss\;(\%)=\frac{w_{0}-w_{t}}{w_{0}}\times 100$$
where *w*_0_ is the dry weight before degradation, and *w_t_* is the dry weight at time *t*.

For 5/5-PTMEG, only 20% weight loss was observed, whereas 5/5-PCL1000 exhibited much faster degradability with 75% weight loss after a week (Figure 10).

#### 3.6.2. Aerobic Degradation Tests

As a result of the biodegradation experiment after 3 months, 68.3 mL (3.42 mmol) of HCl was consumed in the blank test, 44.3 mL (2.22 mmol) in 5/5-PTMEG, and 6.0 mL (0.3 mmol) in 5/5-PCL1000. The amount of HCl consumed was that required to titrate the remaining Ba(OH)_2_ after biodegradation of the film and respiration of microorganisms. Therefore, considering the equivalence ratio, half of the moles of consumed HCl is equal to the number of moles of Ba(OH)_2_.

By subtracting this value from the result of the blank test, the amount of CO_2_ generated as a result of biodegradation and the number of moles of reacted Ba(OH)_2_ could be determined. Because Ba(OH)_2_ and CO_2_ react in a 1:1 ratio, the number of moles of generated CO_2_ is the same as the number of moles of Ba(OH)_2_ used in biodegradation. By multiplying the calculated number of moles of CO_2_ by the MW of CO_2_, the amount of generated CO_2_ through biodegradation can be obtained. The results show that the 5/5-PTMEG sample produced 26.41 mg of CO_2_, whereas the 5/5-PCL1000 sample produced 68.66 mg of CO_2_. The degree of biodegradation was calculated using Equation (4) [51].
(4)$$Degree\;of\;biodegradation\left(\%\right)=\frac{Total\;evoved\;{CO}_{2}-Respired\;{CO}_{2}}{Theoretical\;{CO}_{2}\;value\;from\;the\;sample\;material}\times 100$$

Aromatic polyesters, such as polyethylene terephthalate and polybutylene terephthalate, have excellent mechanical properties but are nearly resistant to microbial attacks, while many aliphatic polyesters are biodegradable but lack mechanical properties compared to aromatic polyester [52,53]. Therefore, the theoretical CO_2_ value from the sample material was calculated assuming that biodegradation occurred only in the SS region and an aliphatic region where biodegradation was likely to occur. In the case of 5/5-PTMEG, the SS region was a form in which PTMEG was combined with DMT through a polycondensation reaction. Therefore, the MW of this region was calculated to be 1132 g mol^−1^, the theoretical number of carbons per molecule was 50, and the theoretical number of CO_2_ molecules was also 50. Accordingly, the possible generation of theoretical CO_2_ was 56 mg (film weight) × 50 (theoretical CO_2_ number) × 44 g mol^−1^ (MW of CO_2_)/1132 g^−1^ mol (MW of SS) = 108.8 mg. In the case of 5/5-PCL1000, the SS region was a form in which PCL polyol was combined with DMT through a polycondensation reaction. Therefore, the MW of this region was calculated to be 1044 g mol^−1^, the theoretical number of carbons per molecule was 181, and the theoretical number of CO_2_ molecules was also 181. Accordingly, the possible generation of theoretical CO_2_ was 57 mg (film weight) × 60 (theoretical CO_2_ number) × 44 g mol^−1^ (MW of CO_2_)/1044 g mol^−1^ (MW of SS) = 144.1 mg. As a result, the degree of biodegradation of 5/5-PTMEG1000 and 4/6-PCL3000 were 24.3% and 47.6%, respectively, demonstrating PCL-based poly(ester ester) TPE shows faster biodegradation than corresponding poly(ether ester) TPE by about two folds.

The degree of biodegradation of two different samples were also qualitatively identified by investigating the surface textures using SEM images (Figure 11) before and after the biodegradation tests. As clearly identified from the films before and after the biodegradation tests, there are lots of signs of degradation on the surface images for PCL-based poly(ester ester) TPE. On the other hand, the surface of poly(ether ester) TPE films remain almost smooth after a month of biodegradation test.

## 4. Conclusions

A series of TPEs was successfully synthesized using telechelic PCL polyols with various MWs as SSs. The α,ω-hydroxyl functionalized PCLs with MWs of 1000–4200 g mol^−1^ and *Ð* of 1.30–1.88 were initially prepared using a DMC catalyst in the presence of a bifunctional EG initiator. DSC and DMA analyses show that PCL-based TPEs have relatively low crystallinity, with *T_g_* ranging from −23 to 0.9 °C, depending on the MW of the PCL SS and the HS/SS ratio. The resultant poly(ester ester) TPEs exhibited excellent tensile strength hardness at room temperature compared with the conventional poly(ether ether) TPE prepared using PTMEG. Accordingly, 5/5-PCL1000 exhibited similar elongation (1104%) but significantly enhanced ultimate strength (33.83 MPa) compared to 5/5-PTMEG1000 (1166% and 23.4 MPa, respectively). The ultimate strength values of the TPEs changed slightly with respect to the MW of the PCL SSs. Among them, 5/5-PCL3000 exhibited the highest ultimate strength (39.72 MPa), whereas a further increase of the MW of SS, 5/5-PCL4000, exhibited a significant decrease in mechanical properties. The crystalline structure and microphase separation morphology of the resultant TPEs were also determined by WAXD and MesoDyn simulation, which showed the characteristic diffraction of the α-form holocrystalline. The domain size of microphase separation of poly(ester ester) TPE tended to be smaller than that of poly(ether ester) TPE. The biodegradability experiments demonstrated that the poly(ester ester) TPE sample, 5/5-PCL1000, was highly degradable with 75% (hydrolytic degradation) and 47.6% (aerobic degradation), much faster than the poly(ether ester) counterpart. In addition to the fundamental investigation of a series of poly(ether ester) TPEs, intensive research on the processing and their practical application as engineering materials must be ongoing research topics.

## Figures and Tables

**Figure 1 polymers-15-03209-f001:**
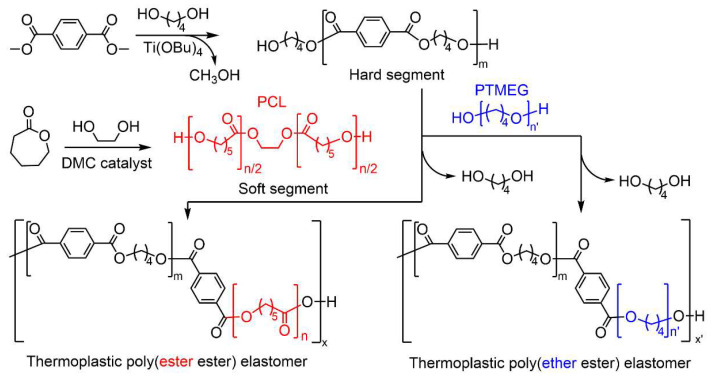
Schematic for the synthesis of thermoplastic poly(ester ester) and poly(ether ester) elastomers using PCL diol and PTMEG soft segments.

**Figure 2 polymers-15-03209-f002:**
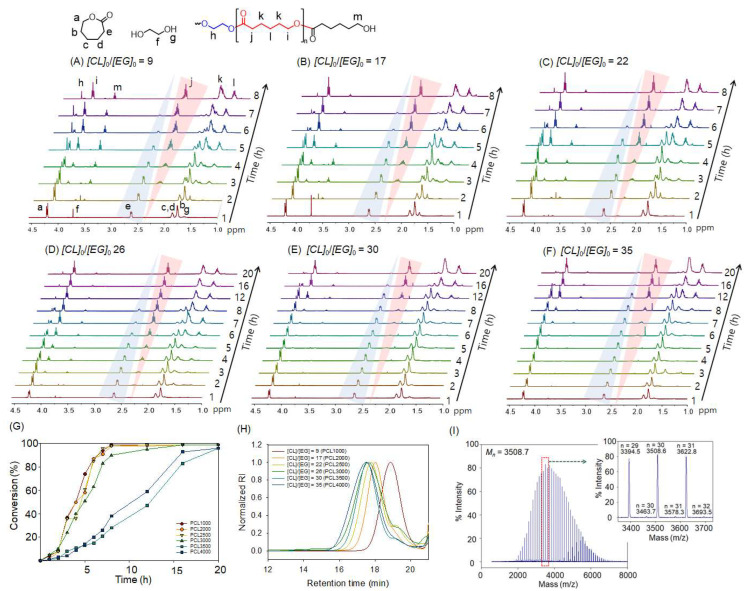
^1^H (400 MHz, CDCl_3_) NMR spectra of reaction mixtures recorded at various intervals polymerized using different initial molar ratios: (**A**) [*CL*]_0_/[*EG*]_0_ = 9 (targeted MW = 1000), (**B**) [*CL*]_0_/[*EG*]_0_ = 17 (targeted MW = 2000), (**C**) [*CL*]_0_/[*EG*]_0_ = 22 (targeted MW = 2500), (**D**) [*CL*]_0_/[*EG*]_0_ = 26 (targeted MW = 3000), (**E**) [*CL*]_0_/[*EG*]_0_ = 30 (targeted MW = 3500), and (**F**) [*CL*]_0_/[*EG*]_0_ = 35 (targeted MW = 4000). (**G**) Monomer conversion versus time curves obtained by ^1^H NMR analysis, (**A**–**F**). (**H**) GPC curves of reaction mixtures of [*CL*]_0_/[*EG*]_0_ = 9, 17, and 22 after 8 h of polymerization, and [*CL*]_0_/[*EG*]_0_ = 26, 30, and 35 after 20 h of polymerization. (**I**) Matrix-assisted laser desorption ionization time of flight mass spectrometry curves of polycaprolactone diol obtained at [*CL*]_0_/[*EG*]_0_ = 26 (PCL3000) (inset is an expanded chromatogram of the marked area).

**Figure 3 polymers-15-03209-f003:**
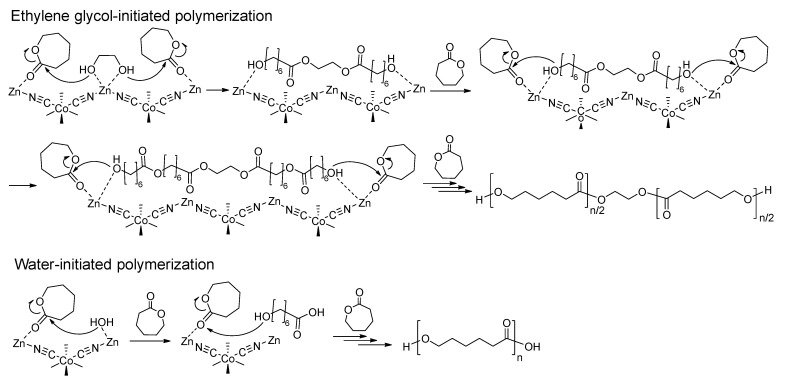
Plausible mechanistic pathways of DMC-catalyzed ROP of CL initiated by EG or H_2_O.

**Figure 4 polymers-15-03209-f004:**
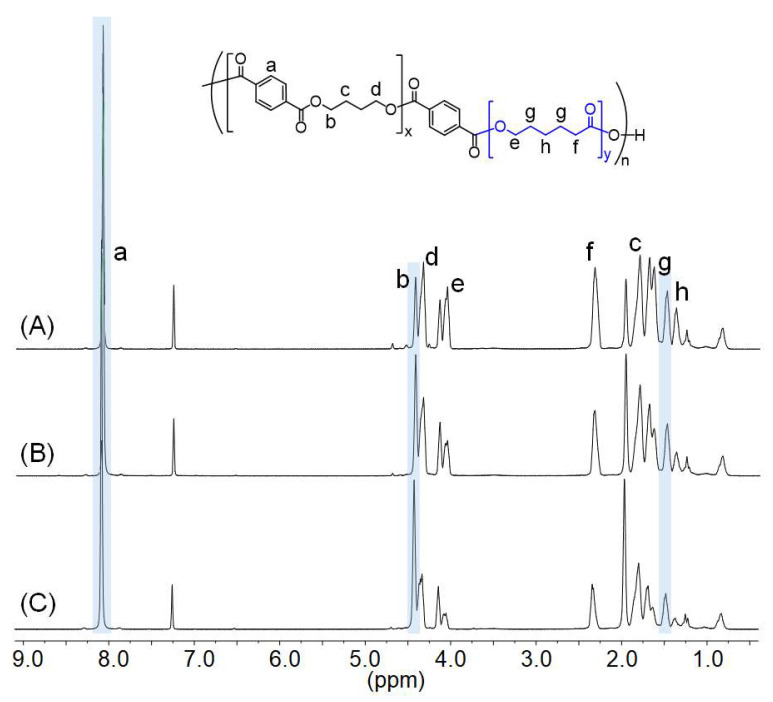
^1^H NMR spectra of thermoplastic poly(ester ester) elastomers prepared using PCL3000 in different HS/SS ratios: (**A**) 40/60, (**B**) 50/50, and (**C**) 60/40. The calculated HS/SS ratios are in Table 3.

**Figure 5 polymers-15-03209-f005:**
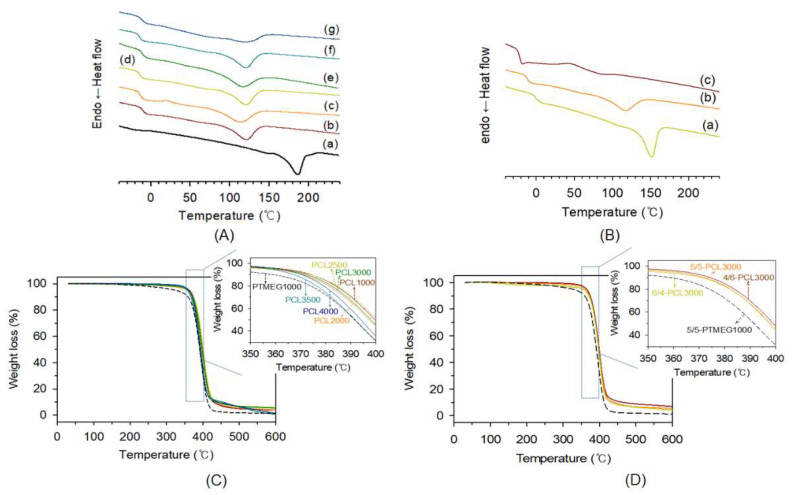
Differential scanning calorimetry curves of thermoplastic elastomers prepared using PTMEG and various polycaprolactone diols with different MWs (**A**): (a) 5/5-PTMEG, (b) 5/5-PCL1000, (c) 5/5-PCL2000, (d) 5/5-PCL2500, (e) 5/5-PCL3000, (f) 5/5-PCL3500, and (g) 5/5-PCL4000, and prepared using various hard to soft segment ratios (**B**): (a) 6/4-PCL3000, (b) 5/5-PCL3000, and (c) 4/6-PCL3000, and thermogravimetric analysis curves of thermoplastic elastomers prepared using PTMEG and various polycaprolactone diols with different MWs (**C**) using various hard to soft segment ratios (**D**).

**Figure 6 polymers-15-03209-f006:**
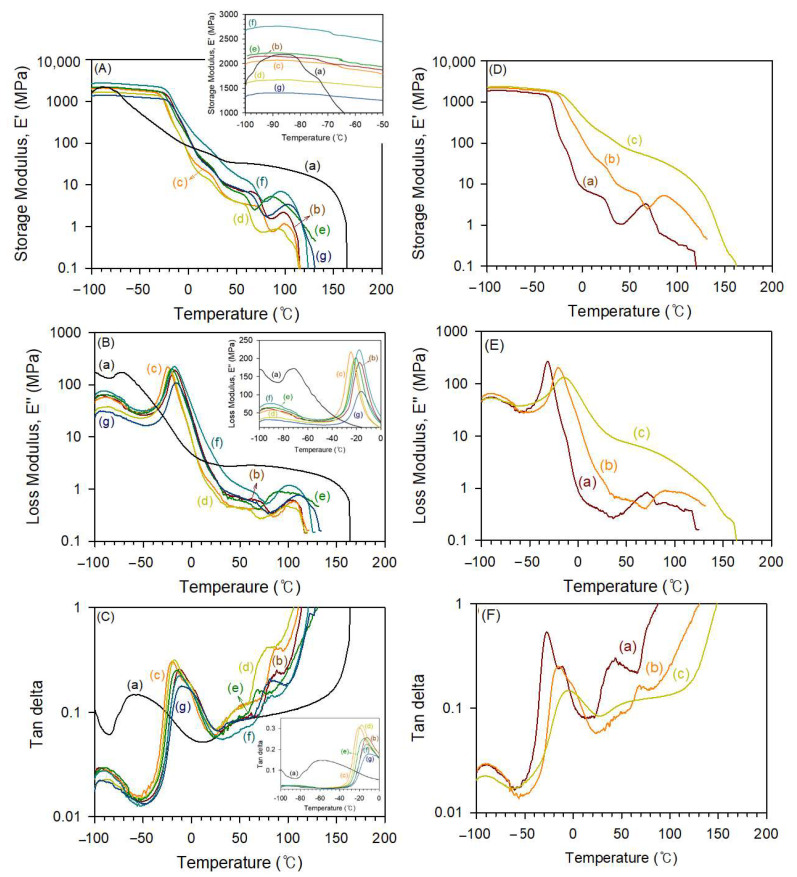
Evolution of elastic modulus *E*′ (**A**,**D**), loss modulus *E*″ (**B**,**E**), and tan *δ* (**C**,**F**) with temperature for thermoplastic elastomers prepared using PTMEG and various polycaprolactone diols with different MWs (**A**–**C**): (a) 5/5-PTMEG, (b) 5/5-PCL1000, (c) 5/5-PCL2000, (d) 5/5-PCL2500, (e) 5/5-PCL3000, (f) 5/5-PCL3500, and (g) 5/5-PCL4000, and prepared using various hard to soft segment ratios (**D**–**F**): (a) 6/4-PCL3000, (b) 5/5-PCL3000, and (c) 4/6-PCL3000.

**Figure 7 polymers-15-03209-f007:**
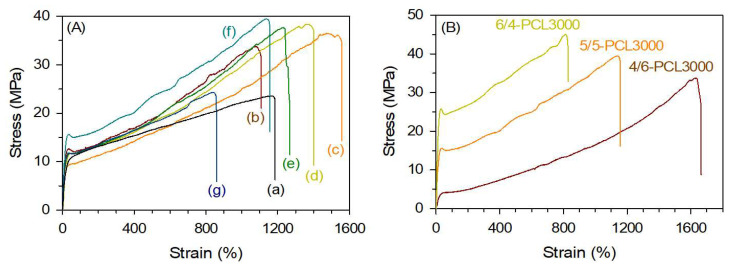
Stress–strain curves of various thermoplastic elastomers prepared using PTMEG and various polycaprolactone diols with different MWs (**A**): (a) 5/5-PTMEG, (b) 5/5-PCL1000, (c) 5/5-PCL2000, (d) 5/5-PCL2500, (e) 5/5-PCL3000, (f) 5/5-PCL3500, and (g) 5/5-PCL4000, and prepared using various hard to soft segment ratios (**B**): 6/4-PCL3000, 5/5-PCL3000, and 4/6-PCL3000.

**Figure 8 polymers-15-03209-f008:**
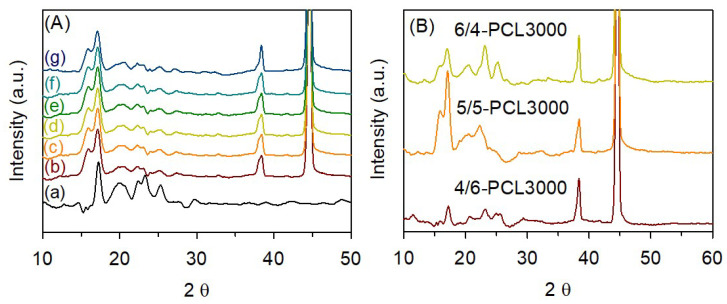
Wide-angle X-ray scattering (WAXS) patterns of various thermoplastic elastomers prepared using PTMEG and various polycaprolactone diols with different MWs (**A**): (a) 5/5-PTMEG, (b) 5/5-PCL1000, (c) 5/5-PCL2000, (d) 5/5-PCL2500, (e) 5/5-PCL3000, (f) 5/5-PCL3500, and (g) 5/5-PCL4000, and prepared using various hard to soft segment ratios (**B**): 6/4-PCL3000, 5/5-PCL3000, and 4/6-PCL3000.

**Figure 9 polymers-15-03209-f009:**
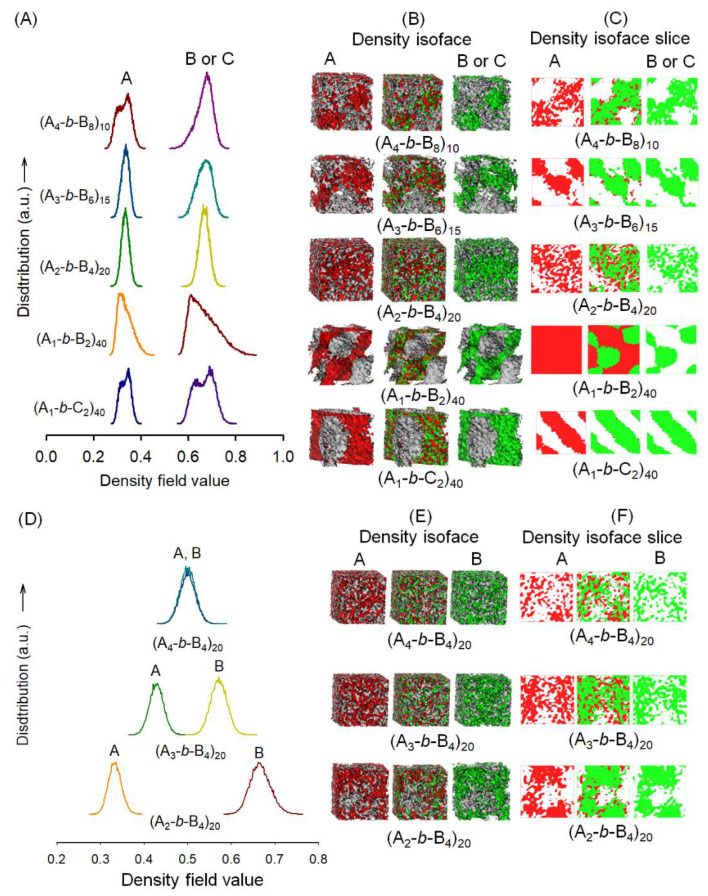
The density field distributions (**A**), beads isosurface representations (**B**), and density slice (**C**) of thermoplastic elastomers prepared using PTMEG or various PCL diols with different MWs after fixing HS/SS = 5/5, where A is HSs consisting of DMT and BDO, B is PCL SSs, and C is PTMEG SSs. The density field distributions (**D**), beads isosurface representations (**E**), and density slice (**F**) of thermoplastic elastomers prepared by modulating hard to soft segment ratios.

**Figure 10 polymers-15-03209-f010:**
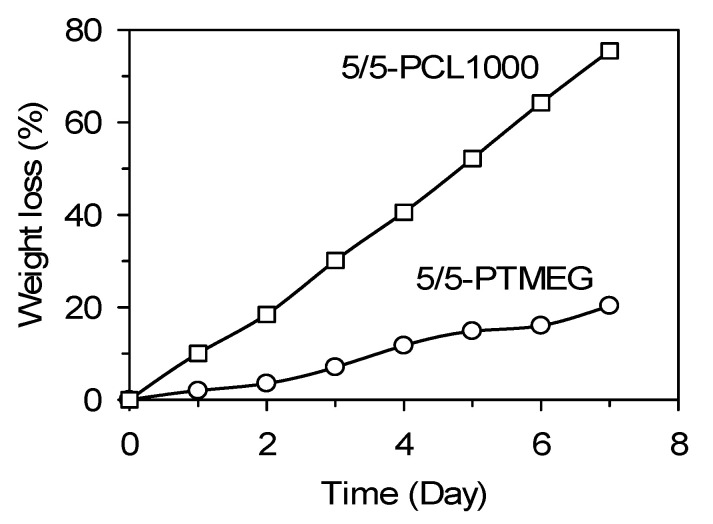
Hydrolysis tests of TPE samples of 5/5-PTMEG and 5/5-PCL1000 at 45 °C for 7 days.

**Figure 11 polymers-15-03209-f011:**
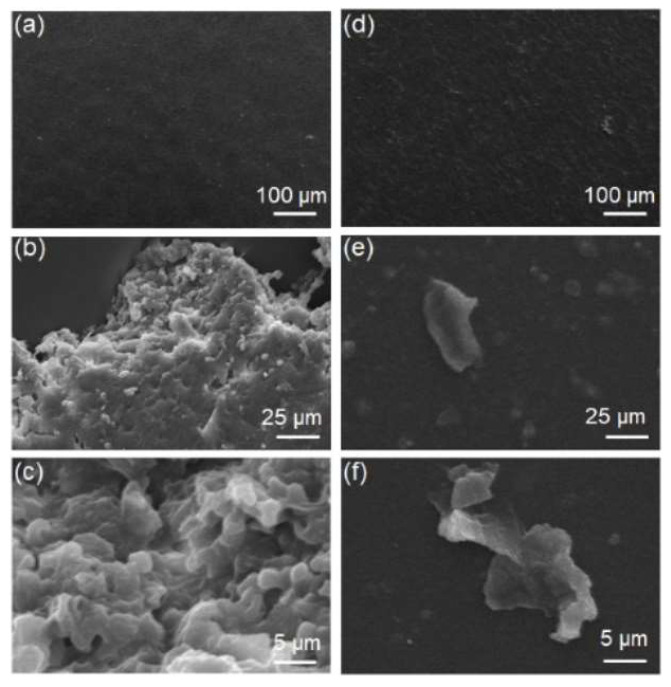
SEM images of 5/5-PCL1000 film (**a**–**c**) and 5/5-PTMEG film (**d**–**f**) before performing biodegradable tests on (**a**,**d**) and after biodegradable tests for a month (**b**,**c**) for 5/5-PCL1000 film and (**e**,**f**) 5/5-PTMEG film.

**Table 1 polymers-15-03209-t001:** Estimated parameters used for the MesoDyn simulation.

Segment	*M_m_* ^a^ (g/mol)	*V_m_* ^b^ (cm^3^/mol)	*δ* ^c^ (J/cm^3^)^0.5^	*C_n_* ^d^	*N_Meso_* ^e^
PCL	114.14	103.61	17.78	5.78	~2
PTMEG	72.11	71.24	18.11	5.99	~2
DMT/BDO	220.23	178.07	19.27	4.15	~1

^a^ Monomer (structural unit) molecular weight. ^b^ Molar volumes at 298 K estimated by Synthia module in Materials Studio^®^. ^c^ van Krevelen solubility parameter estimated by Synthia module. ^d^ Characteristic ratio estimated from Synthia module. ^e^ MesoDyn chain length calculated using: *N_Meso_* = *M_p_*/*M_m_C_n_*, where *M_p_* is the polymer MW assuming 1000.

**Table 2 polymers-15-03209-t002:** Results for the synthesis of PCL diols using a double metal cyanide catalyst and ethylene glycol initiator ^a^.

Sample	[*CL*]_0_*/* [*EG*]_0_	*M_n_*^b^ (g mol^−1^)		*Ð* ^b^	OH value ^c^	*F* ^c^	*T_g_*^d^ (°C)	*T_m_*^d^ (°C)
		Theor.	Measured ^b^					
PCL1000	9	1000	1000	1.3	107.04	1.91	–61.0	31.3/36.3
PCL2000	17	2000	2200	1.4	50.11	1.97	–60.0	44.6/45.9
PCL2500	22	2500	2400	1.5	36.37	1.99	–61.2	46.8/48.7
PCL3000	26	3000	3000	1.5	37.77	2.02	–57.8	45.5/49.5
PCL3500	30	3500	3500	1.6	32.70	2.04	–65.4	49.9/51.1
PCL4000	35	4000	4200	1.8	23.28	1.74	–60.1	50.8/51.9

Abbreviations: *M_n_*, number average molecular weight; *Ð*, polydispersity index; *F*, end functionality; *T_g_*, glass transition temperature; *T_m_*, melting temperature. ^a^ Bulk polymerization conditions: DMC catalyst = 100 mg, temperature = 160 °C, time = 8–20 h, CL = 80 g (0.7 mol), and EG was variable. ^b^ Measured using gel permeation chromatography. ^c^ Measured using a potential difference titrator. ^d^ Measured using differential scanning calorimetry analysis.

**Table 3 polymers-15-03209-t003:** Results of the synthesis of TPEs prepared using PCL diols of different MWs, using PCL3000 in different HS/SS ratios, and using PTMEG.

Sample Code	HS/SS		*T_g_*_1_ ^a^ (°C)	*T_g_*_1_ ^b^ (°C)	*T_g_*_2_ ^b^ (°C)	*T_m_* ^a^ (°C)	*η* (dL g^−1^)	*M_v_* ^c^ (g mol^−1^)	*M_n_* ^d^ (g mol^−1^)
	Feed	NMR							
5/5-PTMEG	50/50	45/55	−22.9	−54.9	42.6	168.9	1.70	113,300	-
5/5-PCL1000	50/50	40/60	−11.1	−11.8	88.3	120.5	1.90	139,500	27,500
5/5-PCL2000	50/50	44/56	−14.4	−20.4	88.0	120.0	1.90	139,600	33,700
5/5-PCL2500	50/50	39/61	−11.6	−17.8	79.0	119.7	2.04	154,800	34,500
5/5-PCL3000	50/50	41/59	−14.7	−15.4	68.3	117.6	2.35	189,400	51,800
5/5-PCL3500	50/50	40/60	−12.8	−12.3	83.2	116.5	1.84	132,600	31,300
5/5-PCL4000	50/50	47/53	−13.9	−10.4	81.5	113.0	1.79	127,500	28,800
4/6-PCL3000	40/60	27/73	−21.1	−27.5	43.8	83.3	1.43	92,800	46,200
6/4-PCL3000	60/40	57/43	−0.5	0.90	48.5	152.2	1.78	132,300	35,300

*T_g_*_1_, Glass transition temperatures of soft phase; *T_g_*_2_, amorphous part in crystalline hard phase; *T_m_*, Melting temperature; *η*, intrinsic viscosity; *M_v_*, viscosity average molecular weight; *M_n_*, number average molecular weight. ^a^ Measured using differential scanning calorimetry. ^b^ Measured using dynamic mechanical analysis. ^c^ Viscosity average molecular weight measured using an Ubbelohde viscometer. ^d^ Measured using gel permeation chromatography.

**Table 4 polymers-15-03209-t004:** Results for the mechanical properties of thermoplastic elastomers.

Sample	HS/SS (wt%)	Ultimate Strength (MPa)	Elongation (%)	Hardness	
				Shore A	Shore D
5/5-PTMEG	50/50	23.42	1166	85	29
5/5-PCL1000	50/50	33.83	1105	90	35
5/5-PCL2000	50/50	36.26	1560	93	38
5/5-PCL2500	50/50	38.46	1395	91	37
5/5-PCL3000	50/50	39.72	1152	93	39
5/5-PCL3500	50/50	37.76	1242	95	39
5/5-PCL4000	50/50	24.47	856	95	40
4/6-PCL3000	40/60	33.96	1638	75	22
6/4-PCL3000	60/40	44.92	820	98	47

## Data Availability

Not applicable.

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
