# Peer review of "Sustainable Polycaprolactone Polyol-Based Thermoplastic Poly(ester ester) Elastomers Showing Superior Mechanical Properties and Biodegradability"

_polymers, 2023, doi:10.3390/polym15153209_

Round 1

Reviewer 1 Report

Dear Authors.

Congratulations on your work which I found interesting.

Manuscript: Sustainable polycaprolactone polyol-based thermoplastic poly(ester ester) elastomers showing superior mechanical properties and biodegradability.

I have some minor revisions to propose to you to improve your work.

Please refer to the following comments.

On page 14 the font should be corrected.

Figures 5 and 6 have too small print screens and also too small descriptions – it is unreadable.

In the „ Thermal properties of TPEs.”

How many runs were made in the DSC test for each of the tested polymers?

Was made a run resetting the history of the tested materials performed?

 In the „Conclusions”

Please indicate specifically in which further research this study can be used - give some examples - Please indicate an additional research plan;

The literature is old, with many citations of works from before 2018 (only 14 items out of 48 are pre-2018) - please consider updating it and extending the scope of the analyzed literature items in line with the subject of the article.

Author Response

Response letter to Reviewer 1’s reports

General comments: I have some minor revisions to propose to you to improve your work. Please refer to the following comments.

Comment 1: On page 14 the font should be corrected.

Response: The font problem will be dissolved during the publication procedure by publisher.

Comment 2: Figures 5 and 6 have too small print screens and also too small descriptions – it is unreadable.

Response: The problems related with the quality of images will be dissolved during the publication procedure. If the publisher asked the high-quality images, we are going to supply them.

Comment 2: In the „ Thermal properties of TPEs.” How many runs were made in the DSC test for each of the tested polymers? Was made a run resetting the history of the tested materials performed?

Response: DSC Heat/Cool/Heat experiments were designed to erase previous thermal history by heating the material above a melt transition, where relaxation or molecular rearrangement can occur, then cooling at a known rate before heating again. The first heating curve provides the “as received” information. The cooling imparts a known thermal history. Therefore, any differences observed between similar materials in the second heating curve are related to real internal differences in the materials (e.g., molecular weight) rather than previous thermal history effects.

Comment 3: In the „Conclusions” Please indicate specifically in which further research this study can be used - give some examples - Please indicate an additional research plan.

Response: As you recommended, we added a sentence at the end of the conclusion part as follows. “In addition to the fundamental investigation of a series of poly(ester ester) TPEs, intensive research on the processing and their practical application as engineering materials must be on-going research topics.”

Comment 4: The literature is old, with many citations of works from before 2018 (only 14 items out of 48 are pre-2018) - please consider updating it and extending the scope of the analyzed literature items in line with the subject of the article.

Response: Thank you for your meaningful comments. I exactly understand what you are worrying about; however, the TPE materials have long history, even though most of them are related with poly(ether ester) TPEs. In fact, there have been no reports to systematically investigate PCL-based poly(ester ester) TPEs, most probably due to the difficulty in synthesizing low MW PCLs. Notwithstanding, we have intensively search for the recent related reports through SciFi and further added them (references 17−20) in the revised manuscript.

  1. Lucherelli, M.A.; Duval, A.; Avérous, L. Biobased vitrimers: Towards sustainable and adaptable performing polymer materials. Prog. Polym. Sci. 2022, 127, 101515, https://doi.org/10.1016/j.progpolymsci.2022.101515.
  2. Thomas, J.; Bouscher, R.F.; Nwosu, J.; Soucek, M.D. Sustainable thermosets and composites based on the epoxides of norbornylized seed oils and biomass fillers. ACS Sustainable Chem. Eng. 2022, 10, 12342−12354, https://doi.org/10.1021/acssuschemeng.2c03434.
  3. Chamas, A.; Moon, H.; Zheng, J.; Qiu, Y.; Tabassum, T.; Jang, J. H.; Abu-Omar, M.; Scott, S. L.; Suh, S. Degradation rates of plastics in the environment. ACS Sustain. Chem. Eng. 2020, 8, 3494−3511, https://doi.org/10.1021/acssuschemeng.9b06635.
  4. Rosenboom, J.G.; Langer, R.; Traverso, G. Bioplastics for a circular economy. Nature Rev. Mater. 7, 117−137, https://doi.org/10.1038/s41578-021-00407-8.

Reviewer 2 Report

1. The writing of formulas is not homogeneous  within the work.

2. The figure 1 must be rewritten.

3. The resolution of the figures must be improved.

4  This reference must be include in the manuscript:

최진혁, & 김일. (2022). Polycaprolactone Polyol-Based Thermoplastic Poly (ester ester) Elastomers showing High-End Properties. 한국고분자학회 학술대회 연구논문 초록집47(1), 74-74.

Author Response

Comment 1. The writing of formulas is not homogeneous within the work.

Response: We have corrected Figure 1 to keep the homogeneity of formulae as the reviewer recommended.

Comment 2. The figure 1 must be rewritten.

Response: We have corrected Figure 1 to keep the homogeneity of formulae as the reviewer recommended.

Comment 3. The resolution of the figures must be improved.

Response: The problems related with the quality of images will be dissolved during the publication procedure. If the publisher asked the high-quality images, we are going to supply them.

Comment 4. This reference must be include in the manuscript:

최진혁, & 김일. (2022). Polycaprolactone Polyol-Based Thermoplastic Poly (ester ester) Elastomers showing High-End Properties. 한국고분자학회 학술대회 연구논문 초록집47(1), 74-74.

Response: We have added the reference in the revised manuscript, reference [44].

Reviewer 3 Report

Manuscript Number: Polymers-2524515

The manuscript written by Choi et al. titled “Sustainable polycaprolactone polyol-based thermoplastic poly(ester ester) elastomers showing superior mechanical properties and biodegradability” provided a comparative study of the use of PCL as SS instead of polyether SS to achieve sustainability and biodegradability. After going through the manuscript, I want to suggest following comments and clarify a few concerns which will further improve the article and make it suitable for Polymers.

1.      The authors have concluded that a double melting of the PCL might be either due to the difference in lamellar thicknesses or the ability of PCL to undergo a thermodynamically favored crystallization state resulting in better crystals. I think that by reducing the cooling and heating rates to 2 °C and calculating the melt enthalpy the abhors will get a better answer. Melt enthalpy should reduce and the peaks should get broadened when the PCL melts will be cooled at a higher cooling rate as melts will vitrify in the amorphous state.

2.       Please improve the quality of the figures.

3.      In terms of tensile properties, an optimum PCL MW of 3500 was noted. I believe that the authors did not put a clear explanation for such behavior. 

Author Response

The manuscript written by Choi et al. titled “Sustainable polycaprolactone polyol-based thermoplastic poly(ester ester) elastomers showing superior mechanical properties and biodegradability” provided a comparative study of the use of PCL as SS instead of polyether SS to achieve sustainability and biodegradability. After going through the manuscript, I want to suggest following comments and clarify a few concerns which will further improve the article and make it suitable for Polymers.

Comment 1. The authors have concluded that a double melting of the PCL might be either due to the difference in lamellar thicknesses or the ability of PCL to undergo a thermodynamically favored crystallization state resulting in better crystals. I think that by reducing the cooling and heating rates to 2 °C and calculating the melt enthalpy the abhors will get a better answer. Melt enthalpy should reduce and the peaks should get broadened when the PCL melts will be cooled at a higher cooling rate as melts will vitrify in the amorphous state.

Response: We sincerely appreciate the reviewer’s valuable comments. As we understand, the crystallization process of semi-crystalline polymers depends on the thermal history of the material. To erase the thermal history, polymers have to be heated to 25–30 °C above their melting temperature. In this way, a homogeneous or isotropic melt is obtained. When the temperatures employed are high enough to produce an isotropic melt, during subsequent cooling in a DSC run, the material will always crystallize at the same temperature. This temperature is usually denoted as the standard crystallization temperature and it only depends on the cooling rate employed. We have carefully designed DSC Heat/Cool/Heat experiments to erase previous thermal history by heating the material above a melt transition, where relaxation or molecular rearrangement can occur, then cooling at a known rate before heating again. The first heating curve provides the “as received” information. The cooling imparts a known thermal history. Therefore, any differences observed between similar materials in the second heating curve are related to real internal differences in the materials (e.g., molecular weight) rather than previous thermal history effects. Through independent tests by reducing the cooling and heating rates, we still observed a double melting of the PCL with slightly more narrow peaks.

Comment 2. Please improve the quality of the figures.

Response: The problems related with the quality of images will be dissolved during the publication procedure. If the publisher asked the high-quality images, we are going to supply them.

Comment 3. In terms of tensile properties, an optimum PCL MW of 3500 was noted. I believe that the authors did not put a clear explanation for such behavior. 

Response: As the reviewer commented, PCL MW of 3500 showed the best tensile properties (Figure 7). There must be lots of possible reasons to show this behavior beyond the experimental error. However, it may be too premature to conclude only with the collected data in this study, and must be one of the on-going research topics. We appreciate the reviewer’s keen comments.

Reviewer 4 Report

The study of the Sustainable polycaprolactone polyol-based thermoplastic poly(ester ester) elastomers showing superior mechanical properties and biodegradability warrants as good research. The research is particularly interesting with the vast majority of polymer sustainability research happening in elastomers and products in general which are of utmost importance.

In the introduction, authors talk about sustainaibilty, it will be beneficial if you add references of the recent review articles on polymer products sustainability by Luc Averous, Jomin thomas,  Jan-Georg Rosenboom etc who has talked about bio-based polymers for different applications.

However, figure quality needs to be significantly improved. The data makes sense but its hard to read the figures. 

In totality, the paper is well written and need only minor changes. TPE elastomers is also a topic that’s of huge importance to the industry and the work does shed a good among of knowledge onto them.  

Good

Author Response

The study of the Sustainable polycaprolactone polyol-based thermoplastic poly(ester ester) elastomers showing superior mechanical properties and biodegradability warrants as good research. The research is particularly interesting with the vast majority of polymer sustainability research happening in elastomers and products in general which are of utmost importance.

Comment 1. In the introduction, authors talk about sustainaibilty, it will be beneficial if you add references of the recent review articles on polymer products sustainability by Luc Averous, Jomin thomas,  Jan-Georg Rosenboom etc who has talked about bio-based polymers for different applications.

Response: We have added references 17−20 in the revised manuscript, as the reviewer recommended for readers’ convenience:

  1. Lucherelli, M.A.; Duval, A.; Avérous, L. Biobased vitrimers: Towards sustainable and adaptable performing polymer materials. Prog. Polym. Sci. 2022, 127, 101515, https://doi.org/10.1016/j.progpolymsci.2022.101515.
  2. Thomas, J.; Bouscher, R.F.; Nwosu, J.; Soucek, M.D. Sustainable thermosets and composites based on the epoxides of norbornylized seed oils and biomass fillers. ACS Sustainable Chem. Eng. 2022, 10, 12342−12354, https://doi.org/10.1021/acssuschemeng.2c03434.
  3. Chamas, A.; Moon, H.; Zheng, J.; Qiu, Y.; Tabassum, T.; Jang, J. H.; Abu-Omar, M.; Scott, S. L.; Suh, S. Degradation rates of plastics in the environment. ACS Sustain. Chem. Eng. 2020, 8, 3494−3511, https://doi.org/10.1021/acssuschemeng.9b06635.
  4. Rosenboom, J.G.; Langer, R.; Traverso, G. Bioplastics for a circular economy. Nature Rev. Mater. 7, 117−137, https://doi.org/10.1038/s41578-021-00407-8.

Comment 2. However, figure quality needs to be significantly improved. The data makes sense but its hard to read the figures. 

Response: The problems related with the quality of images will be dissolved during the publication procedure. If the publisher asked the high-quality images, we are going to supply them.

Comment 3. In totality, the paper is well written and need only minor changes. TPE elastomers is also a topic that’s of huge importance to the industry and the work does shed a good among of knowledge onto them.  

Response: We appreciate the reviewer’s valuable comments.